# The Genetic and Molecular Analyses of *RAD51C* and *RAD51D* Identifies Rare Variants Implicated in Hereditary Ovarian Cancer from a Genetically Unique Population

**DOI:** 10.3390/cancers14092251

**Published:** 2022-04-30

**Authors:** Wejdan M. Alenezi, Larissa Milano, Caitlin T. Fierheller, Corinne Serruya, Timothée Revil, Kathleen K. Oros, Supriya Behl, Suzanna L. Arcand, Porangana Nayar, Dan Spiegelman, Simon Gravel, Anne-Marie Mes-Masson, Diane Provencher, William D. Foulkes, Zaki El Haffaf, Guy Rouleau, Luigi Bouchard, Celia M. T. Greenwood, Jean-Yves Masson, Jiannis Ragoussis, Patricia N. Tonin

**Affiliations:** 1Department of Human Genetics, McGill University, Montreal, QC H3A0C7, Canada; wagdan.alenizy@mail.mcgill.ca (W.M.A.); caitlin.fierheller@mail.mcgill.ca (C.T.F.); timothee.revil@mcgill.ca (T.R.); supriya.behl@mail.mcgill.ca (S.B.); dan.spiegelman@mcgill.ca (D.S.); simon.gravel@mcgill.ca (S.G.); william.foulkes@mcgill.ca (W.D.F.); guy.rouleau@mcgill.ca (G.R.); celia.greenwood@mcgill.ca (C.M.T.G.); ioannis.ragoussis@mcgill.ca (J.R.); 2Cancer Research Program, Centre for Translational Biology, The Research Institute of McGill University Health Centre, Montreal, QC H4A3J1, Canada; corinne.serruya@affiliate.mcgill.ca (C.S.); suzanna.arcand@mail.mcgill.ca (S.L.A.); porangana.nayar@mail.mcgill.ca (P.N.); 3Department of Medical Laboratory Technology, Taibah University, Medina 42353, Saudi Arabia; 4Department of Molecular Biology, Medical Biochemistry and Pathology, Laval University, Quebec City, QC G1V0A6, Canada; larissa.milano-de-souza@crchudequebec.ulaval.ca (L.M.); jean-yves.masson@crchudequebec.ulaval.ca (J.-Y.M.); 5Genome Stability Laboratory, Centre de Recherche du Centre Hospitalier de l’Université de Québec, HDQ Pavilion, Oncology Axis, Quebec City, QC G1R2J6, Canada; 6McGill Genome Centre, McGill University, Montreal, QC H3A0G1, Canada; 7Lady Davis Institute for Medical Research, Jewish General Hospital, Montreal, QC H3T1E2, Canada; kathleen.klein@mail.mcgill.ca; 8Institute of Parasitology, McGill University, Montreal, QC H9X3V9, Canada; 9Montreal Neurological Institute, McGill University, Montreal, QC H3A2B4, Canada; 10Institut du Cancer de Montréal, Centre de Recherche du Centre Hospitalier de l’Université de Montréal, Montreal, QC H2X0A9, Canada; anne-marie.mes-masson@umontreal.ca (A.-M.M.-M.); diane.provencher.med@ssss.gouv.qc.ca (D.P.); ahmed.zaki.anwar.el.haffaf.med@ssss.gouv.qc.ca (Z.E.H.); 11Département de Médecine, Université de Montréal, Montreal, QC H3T1J4, Canada; 12Division of Gynecologic Oncology, Université de Montréal, Montreal, QC H4A3J1, Canada; 13Department of Medical Genetics, McGill University Health Centre, Montreal, QC H4A3J1, Canada; 14Department of Medicine, McGill University, Montreal, QC H3G2M1, Canada; 15Gerald Bronfman Department of Oncology, McGill University, Montreal, QC H3A1G5, Canada; 16Service de Médecine Geénique, Centre Hospitalier de l’Université de Montréal, Montreal, QC H2X0A9, Canada; 17Department of Biochemistry and Functional Genomics, Université de Sherbrooke, Sherbrooke, QC J1K2R1, Canada; luigi.bouchard@usherbrooke.ca; 18Department of Medical Biology, Centres Intégrés Universitaires de Santé et de Services Sociaux du Saguenay-Lac-Saint-Jean Hôpital Universitaire de Chicoutimi, Saguenay, QC G7H7K9, Canada; 19Centre de Recherche du Centre Hospitalier l’Université de Sherbrooke, Sherbrooke, QC J1H5N4, Canada; 20Department of Epidemiology, Biostatistics and Occupational Health, McGill University, Montreal, QC H3A1A2, Canada

**Keywords:** *RAD51C*, *RAD51D*, ovarian cancer predisposing genes, French Canadian, genetic drift

## Abstract

**Simple Summary:**

We have investigated *RAD51C* and *RAD51D*, hereditary ovarian cancer risk genes, in French Canadians of Quebec, Canada. This population of Western European origins exhibits a unique genetic landscape as shown by the frequency of carriers of specific rare pathogenic variants. As studying French Canadians could facilitate the identification and interpretation of clinically relevant variants, we performed genetic analyses of *RAD51C* and *RAD51D* in this population comprised of cases with a family history of ovarian cancer or those who developed it at a young age. We identified candidate variants and then investigated them in other French Canadian study groups. We performed biological assays and revealed possible mechanisms that would affect gene function. Using engineered cells expressing one of our protein variants, we also showed that they were more sensitive to a recently approved treatment for ovarian cancer. Our findings support the role of inherited variants in *RAD51C* and *RAD51D* in ovarian cancer.

**Abstract:**

To identify candidate variants in *RAD51C* and *RAD51D* ovarian cancer (OC) predisposing genes by investigating French Canadians (FC) exhibiting unique genetic architecture. Candidates were identified by whole exome sequencing analysis of 17 OC families and 53 early-onset OC cases. Carrier frequencies were determined by the genetic analysis of 100 OC or HBOC families, 438 sporadic OC cases and 1025 controls. Variants of unknown function were assayed for their biological impact and/or cellular sensitivity to olaparib. *RAD51C* c.414G>C;p.Leu138Phe and c.705G>T;p.Lys235Asn and *RAD51D* c.137C>G;p.Ser46Cys, c.620C>T;p.Ser207Leu and c.694C>T;p.Arg232Ter were identified in 17.6% of families and 11.3% of early-onset cases. The highest carrier frequency was observed in OC families (1/44, 2.3%) and sporadic cases (15/438, 3.4%) harbouring *RAD51D* c.620C>T versus controls (1/1025, 0.1%). Carriers of c.620C>T (*n* = 7), c.705G>T (*n* = 2) and c.137C>G (*n* = 1) were identified in another 538 FC OC cases. *RAD51C* c.705G>T affected splicing by skipping exon four, while RAD51D p.Ser46Cys affected protein stability and conferred olaparib sensitivity. Genetic and functional assays implicate *RAD51C* c.705G>T and *RAD51D* c.137C>G as likely pathogenic variants in OC. The high carrier frequency of *RAD51D* c.620C>T in FC OC cases validates previous findings. Our findings further support the role of *RAD51C* and *RAD51D* in hereditary OC.

## 1. Introduction

Ovarian cancer (OC) has a high estimated heritable component (39%, 95% confidence interval [CI]: 23–55) [1]. Harbouring loss-of-function (LoF) variants in *BRCA1* [2] or *BRCA2* [3] confers significant lifetime risk of developing OC, which accounts for 40–85% of OC cases in hereditary breast and ovarian cancer (HBOC) syndrome families and 10–15% of those not selected for age at OC diagnosis and/or family history of cancer [4]. Carrying rare LoF variants in *RAD51C* [5] and *RAD51D* [6] has been associated with OC predisposition in different populations [7], though the penetrance has yet to be determined. These genes encode RAD51 paralog proteins that are structurally similar to the RAD51 recombinase, which allows faithful DNA double-strand break repair along with BRCA1 and BRCA2 by the homologous recombination (HR) pathway [8]. RAD51C [9] and RAD51D [6] deficient cells or those expressing pathogenic variants in these genes [6,10] have been shown to render sensitivity to poly (ADP-ribose) polymerase (PARP) inhibitors such as olaparib, which is the first to be approved for OC treatment [11]. Indeed, four PARP inhibitors are currently approved for clinical use: olaparib, rucaparib, niraparib, and talazoparib for the treatment of *BRCA1* and *BRCA2* pathogenic variant-positive OC, breast, pancreatic and prostate cancers (recently reviewed [11]). PARP inhibitors, including olaparib, have been proven effective in the treatment of OC in the context of harbouring pathogenic *BRCA1* and *BRCA2* variants [12,13,14,15,16]. Thus, identifying carriers of pathogenic variants in *RAD51C* and *RAD51D* might be useful for identifying those who might benefit from management of OC with protocols using novel PARP inhibitors. Less than 2% of OC cases have been reported to harbour LoF pathogenic variants in *RAD51C* or *RAD51D* [7,17,18,19]. However, the role of pathogenic rare missense variants in these genes has not been fully explored, although approximately 39% are predicted to be damaging by in silico tools [17].

Investigating populations exhibiting unique genetic architecture due to common ancestors has the potential to facilitate the characterization of pathogenic variants in known or candidate cancer predisposing genes [20,21,22]. Our research on the French Canadians (FC) of Quebec (Canada), has shown that a small number of specific pathogenic variants account for the majority of *BRCA1* or *BRCA2* implicated HBOC and hereditary breast cancer (HBC) syndrome families, whereas a vast spectrum of variants in these genes has been reported in the general population [21,23,24]. Likewise, only one pathogenic variant accounts for all *PALB2* implicated FC HBC syndrome families [21,25,26]. The unique genetic architecture of FCs has been attributed to genetic drift as result of multiple waves of localized population expansion in Quebec of European (France) settlers since 1608 [20,21,27]. Given the expectation that a higher frequency of carriers of rare variants would be observed in cancer cases versus controls in populations exhibiting genetic drift, the genetic analyses of FCs has the potential to identify clinically relevant pathogenic variants in new risk genes [20,21,22]. For example, recently, we reported that *RAD51D* c.620C>T; p.Ser207Leu, initially identified by genetic panel testing of familial OC cases in clinical settings, was found to occur at a significantly higher frequency in FC OC cases versus controls [28]. At the time of discovery, this *RAD51D* variant was classified as a missense variant of unknown clinical significance (VUS). Its classification has since been revised to pathogenic or likely pathogenic, especially as the same study reported that *RAD51D* p.Ser207Leu impaired HR function and rendered cells sensitive to olaparib. Although the role of *RAD51C* and *RAD51D* have yet to be fully explored in the FC population [21,29,30], it is clear from previous work that investigating the FC population can assist in characterizing new cancer risk genes (reviewed in [21]).

The main aim of this study was to identify and investigate candidate variants in *RAD51C* and *RAD51D* in FCs with OC by: (1) performing whole exome sequencing (WES) and bioinformatic analyses of the germline of familial and sporadic early-onset OC cases; (2) determining the carrier (harbouring the variant in the heterozygous state) frequencies of candidate variants by genotyping or surveying available genetic data in OC cases and population-matched controls; (3) assaying available tumour DNA from carriers for loss of heterozygosity (LOH) of *RAD51C* and *RAD51D* loci; (4) describing associated clinico-pathological features of OC in carriers; and (5) using biological assays involving cancer cell line models to assess the impact of missense variants with unknown biological function, including sensitivity to PARP inhibitors.

## 2. Materials and Methods

### 2.1. FC Study Participants

The FC participants investigated in this study were selected from independently established biobank resources as described in Appendix A. All participants were recruited independently to these biobanks in accordance with ethical guidelines of their respective Institutions Research Ethics Boards, including those participants from adult hereditary cancer clinics in Quebec. The participants that had provided their associated biological specimens (DNA), genetic data, pedigree information and clinical metrics, where appropriate, were anonymized at source by the respective biobanks. To further protect the anonymity of study subjects, all samples were assigned a unique identifier and pedigrees modified. This project was conducted with approval and in accordance with the guidelines of The McGill University Health Centre Research Ethics Board (MP-37-2019-4783).

As described in Figure 1A, candidate variants in *RAD51C* and *RAD51D* were discovered (phase I) in peripheral blood lymphocyte (PBL) DNA from 20 familial (from 17 OC families) and 53 sporadic early-onset OC cases known to be negative for pathogenic *BRCA1* and *BRCA2* variants by FC mutation-panel or clinical multi-gene panel testing from information provided by adult hereditary cancer clinics in Quebec or the Banque de tissus et données of the Réseau de recherche sur le cancer of the Fond de recherche du Québec—Santé (RRCancer biobank) (rrcancer.ca).

Carrier frequencies of *RAD51C* and *RAD51D* candidate variants were determined (phase II) by genotyping PBL DNA from index cancer cases from three different FC cancer groups: 44 OC and 56 HBOC families, and 438 sporadic OC cases regardless of the carrier status of *BRCA1* and *BRCA2* pathogenic variants (RRCancer biobank). Carrier frequencies were also determined by investigating genetic data available from population-matched controls from three established biobanks: (1) CARTaGENE (cartagene.qc.ca) [31]; (2) McGill University—Montreal Neurological Institute (MNI) [28,32]; and (3) Sherbrook University—glucose regulation in gestation and growth (Gen3G) [33].

To increase the pool of carriers of our candidate variants (phase III), we genotyped another 538 PBL or tumour DNA samples from recently available OC cases that were provided by the RRCancer biobank.

Age at diagnosis, tumour grade, stage of the disease, histopathology subtype as confirmed by a gynecologic pathologist, personal history of other cancers, chemotherapy treatment and/or overall outcome were provided for selected OC cases from the RRCancer biobank if available. For comparative purposes, clinical data was also provided by the RRCancer biobank from carriers of the pathogenic *BRCA1* c.4327C>T; p.Arg1443Ter from the sporadic OC case study group, investigated previously by our group for *BRCA1* and *BRCA2* carrier status [34].

All study groups described in this report were selected from participants that had been recruited from the province of Quebec to participate in various biobanking projects: familial OC or invasive breast cancer (BC) cases self-reported FC ancestry [23,24]; the majority of sporadic OC (at least 88%) self-reported FC ancestry [34]; additional OC cases self-reported FC ancestry by RRCancer biobank; Gen3G project included mothers that self-reported FC ancestry [33]; MNI controls were self-reported as FC ancestry [28,32]; and CARTaGENE resource defined FC status of controls based on being self-reported as FC, born in Quebec, having parents and all four grandparents born in Canada and having French as first language learned [31].

### 2.2. Identification and Verification of Candidate Variants

To identify candidate variants, PBL DNA from a total of 73 OC phase I cases (Figure 1A) were subjected to WES at the McGill Genome Centre using NimbleGen SeqCap EZ Exome v3.0 library kit (Roche, NJ, USA), followed by paired-end sequencing of 150 bp reads on different Illumina HiSeq platforms. Reads were aligned to the human reference genome assembly GRCh37/hg19 using Burrows-Wheeler aligner v0.7.17, followed by PCR deduplication using Picard v2.9.0. Realignment around small insertions and deletions was performed and then, germline variants were called using HaplotypeCaller using Genome Analysis Toolkit (GATK) v3.5. Variants were then filtered for base sequencing quality score ≥ 30 and annotated using Ensembl Variant Effect Predictor (VEP) and GEMINI v0.19.1.

Variants in *RAD51C* and *RAD51D* were extracted from the annotated variant call files (VCF) for filtering and prioritization (Figure 1B). Silent and intronic variants and those with a minor allele frequency (MAF) ≥ 0.01 in the Genome Aggregation Database of non-cancer population (gnomAD) v2.1.1 (gnomad.broadinstitute.org) [35,36], low coverage (<10 reads) and variant allele frequency (VAF) <0.20 and >0.80 were filtered out [37]. Manual inspection was performed to confirm variants using Integrative Genomics Viewer (IGV) v2.4.10.

We prioritized for further investigation (see Figure 1B) LoF variants and missense variants classified as pathogenic, likely-pathogenic or VUS in ClinVar [38,39] and the American College of Medical Genetics and Genomics (ACMG) guidelines [40]. We prioritized variants predicted to overlap conserved residues or be damaging at the RNA or protein level by at least one of several in silico tools selected based on their best performance [41,42,43]. Briefly, we selected variants having a conserved prediction score ≥ 2.0 by Genomic Evolutionary Rate Profiling v1.0 (GERP++ v1.0) [44], ≥0.2 Phylogenetic *p* value v4.0 of 100 vertebrates (PhyloP 100 way v4.0) and ≥0.4 by PHAST Conservation v4.0 of 100 vertebrates (PhastCons 100 way v4.0) [45]; damaging prediction score ≥ 0.4 by Rare Exome Variant Ensemble Learner v4.0 (REVEL v4.0) [46], Meta-analytic Logistic Regression v4.0 (MetaLR v4.0) [47] and CONsensus DELeteriousness v2.0 (CONDEL v2.0) [48], ≥0.0 by Meta-analytic Support Vector Machine v4.0 (MetaSVM v4.0) [47]; ≥−2.5 by Protein Variation Effect Analyzer v4.0 (PROVEAN v4.0) [49] and ≥20 (Phred score = −10 log10 P) by Combined Annotation Dependent Depletion v1.4 (CADD v1.4) [50]. Prediction performance of these in silico tools was tested based on germline variants submitted to the ClinVar database and classified by ACMG guidelines [41,43]. Variants were predicted to affect splicing if the score was ≥0.4 by two of Database Splicing Consensus Single Nucleotide Variant (dbscSNV) in silico tools [45]: AdaBoost v1.1 (ADA v1.1) or Random Forest v1.1 (RF v1.1); or classified to affect splicing by Maximum Entropy Estimates of Splice Junction Strengths v2.0 (MaxEntScan v2.0) [51] and Human Splicing Finder v3.1 (HSF v3.1) [52]. Prediction performance of these splicing predictor in silico tools was tested on somatic variants submitted to the catalogue of somatic mutations in cancer (COSMIC) database [42].

Candidate variants were verified in PBL DNA by bidirectional Sanger sequencing of PCR products using customized primers (Appendix A) at the McGill Genome Centre as previously described [28]. Sequencing chromatograms were visually inspected for variant heterozygosity using 4Peaks v1.8. (nucleobytes.com/4peaks/) (The Netherlands Cancer institute, Amsterdam, The Netherlands).

### 2.3. Investigating Carrier Frequencies of Candidate Variants in FC OC Cases and Controls

Carrier frequencies of candidate variants were determined by genotyping PBL DNA from OC or BC cases from defined FC study groups (Appendix A) using customized TaqMan assays [53] (Appendix A). Carriers were verified by bidirectional Sanger sequencing as described above. Corresponding tumour DNA from the index case was genotyped where PBL DNA was not available.

Carrier frequencies of candidate variants were determined in FC controls by surveying 1025 sequencing-based: 433 from Gen3G, 422 from MNI and 170 from CARTaGENE [28,31,33] and 8493 SNP genotyping-based [31] data available from CARTaGENE resources (Appendix A). Carrier counts were extracted from VCF files or PLINK files. If the variant probes were not represented on the SNP array, pre-phasing and imputation were performed using Eagle2 with the Burrows-Wheeler transformation through Sanger Imputation Services (sanger.ac.uk/tool/sanger-imputation-service/, accessed on 1 February 2019), where Haplotype Reference Consortium release v1.1 (HRC.r1) was used as the reference [54,55].

Fisher’s exact test was used to compare carrier or allele frequencies in appropriate OC study groups versus controls using two-tailed *p* values where <0.05 was considered significant. A permutation analysis using Fisher’s exact test was also performed of cases and controls to address the possibility that carriers could have been recruited to more than one FC study group (Appendix A).

### 2.4. Surveying Allele and Carrier Frequencies of Candidate Variants in Genetic Databases of Non-FC Populations

Carrier or allele frequencies of candidate variants were surveyed in genetic data that was derived from cancer-free individuals of European ancestry, as the original founders of FCs of Quebec were of Western European (France) origin [20,21,27,56]. Carrier or allele frequencies were determined by querying data available from three resources (Appendix A). Allele frequencies for the non-cancer of non-Finnish European ancestry were extracted from WES or whole genome sequencing (WGS) data deposited in the Genome Aggregation Database (gnomAD) v2.1.1 (gnomad.broadinstitute.org, accessed on 1 October 2021). For comparative purposes, we also extracted data derived from other populations from this resource. Carrier frequencies for 7325 women of European ancestry regardless of family history of cancer were extracted from genetic data from the Fabulous Ladies Over Seventy (FLOSSIES) project (whi.color.com, accessed on 1 October 2021), which included data from a 27-gene panel sequencing study.

Candidate variants were also queried in genetic data from 25,509 OC cases and 40,491 controls using the Ovarian Cancer Association Consortium (OCAC) database (ocac.ccge.medschl.cam.ac.uk/data-projects/, accessed on 15 June 2020). Summary statistics provided included odd ratios (OR_Log2_) with *p* values comparing all epithelial OC histopathological subtypes with OCAC controls.

For comparison purposes, we also queried *BRCA1* c.4327C>T; p.Arg1443Ter in all of these resources, as this variant is the most frequently reported pathogenic *BRCA1* variant in the FC and European populations [21].

### 2.5. LOH Analysis of RAD51C and RAD51D Loci in OC Tumour DNA from Candidate Variant Carriers

LOH analysis of *RAD51C* and *RAD51D* loci was performed by Sanger sequencing of OC tumour DNA from variant carriers, where possible, using customized primers as described above (Appendix A). Extracted DNA from fresh-frozen (FF) or histopathological sections from formalin-fixed paraffin-embedded (FFPE) tumour tissues were provided from the RRCancer biobank for DNA extraction based on the manufacturer’s instructions (Promega, Canada). Sequencing chromatograms were visually inspected for complete or partial loss of the wild-type *RAD51C* or *RAD51D* alleles from carriers using 4Peaks v1.8. (nucleobytes.com/4peaks/) (The Netherlands Cancer institute, Amsterdam, The Netherlands).

### 2.6. RNA Extraction and Reverse Transcription Analyses of RAD51C

An Epstein–Barr virus transformed lymphoblastoid cell line (LCL) was established from PBLs from *RAD51C* c.705G>T carriers and non-carriers of this variant as previously described [57]. Approximately 5,000,000 cells were treated with 28 µg/mL of cycloheximide or DMSO for 3 h. RNA was extracted from cell pellets treated with 1000 µL of TRIzol (Invitrogen, Canada) for reverse transcription [58]. cDNA was amplified and purified for Sanger sequencing using customized primers as described above (Appendix A). Sequencing chromatograms were visually inspected for splicing impact using 4Peaks v1.8. (nucleobytes.com/4peaks/) (The Netherlands Cancer institute, Amsterdam, The Netherlands).

### 2.7. Cell Lines

We used three cell lines in our assays, HeLa (cervical carcinoma), U2OS (sarcoma) and OVCAR-3 (epithelial ovarian adenocarcinoma). HeLa cells were obtained from American Type Culture Collection (atcc.org) and maintained in Dulbecco’s Modified Eagle Media supplemented with 10% Fetal Bovine Serum and 1% Penicillin/Streptomycin, at 37 °C, 5% CO_2_, and 20% O_2_. OVCAR-3 cells were grown in RPMI supplemented with 0.01 mg/mL bovine insulin and 20% foetal bovine serum, at 37 °C, 5% CO_2._ U2OS cells RAD51D knock-out (KO) [59] were purchased from the Deutsche Sammlung von Mikroorganismen und Zellkulturen GmbH (U2OS#18-RAD51D-4, DSMZ, Braunschweig, Germany) maintained in Dulbecco’s Modified Eagle Media supplemented with 10% Fetal Bovine Serum and 1% Penicillin/Streptomycin, at 37  °C, 5% CO_2_, and 20% O_2_. OVCAR-3 cells were a kind gift from Dimcho Batchvarov (CHU de Québec).

### 2.8. Complementation Assays and siRNA Transfections

For in cellulo experiments in HeLa and OVCAR-3 cells, the RAD51D p.Ser46Cys protein variant was obtained via site-directed mutagenesis using the Q5 Site-Directed Mutagenesis Kit (NEB) (New England biolabs, Canada) using the pcDNA3-RAD51D wild-type (WT) as a template and primers listed on Appendix A. The mammalian expression vector pcDNA3-RAD51D that was used as a template had been previously modified for RAD51D expression with a FLAG tag at the N-terminus and for resistance to RAD51D siRNA with the Q5 Site-Directed Mutagenesis Kit (NEB) and primers listed on Appendix A. The siRNA target sequence used to silence RAD51D was siRAD51D #7 GGCCAAAUCUUCCCGACAGdTdT and the non-specific siRNA used as control was UUCGAACGUGUCACGUCAAdTdT.

Approximately 240,000 cells were seeded into one well of a six-well plate before being double transfected 24 and 48 h later with 50 nM control or RAD51D siRNA using Lipofectamine RNAiMAX (Invitrogen, Canada). For Hela, cells were then complemented by transfection of 800 ng of the pcDNA3 empty vector or the indicated siRNA-resistant FLAG-tagged RAD51D construct using Lipofectamine 2000 (Invitrogen, Canada) for four hours. Transient transfections in OVCAR3 cells were performed with 300 or 700 ng of the pcDNA3 empty vector or the indicated siRNA-resistant FLAG-tagged RAD51D construct using Lipofectamine 3000 (Invitrogen, Canada) for four hours according to the manufacturer’s protocol. U2OS RAD51D KO cells were stably complemented using the AAVS1 integration system [60]. The AAVS1 RAD51D WT or p.Ser46Cys constructs were generated by amplification using the pcDNA3 plasmids previously described as a template and primers listed on Appendix A; both products were cloned into the AAVS1 vector in NotI/PspXI sites. Briefly, cells were transfected with the 4 μg of the AAVS1 construct containing either the WT or the RAD51D p.Ser46Cys variant, along with the 0.4 μg of the pZFN plasmid, using Lipofectamine 2000 (Invitrogen) for 4 h. 24 h later, transfected cells were selected with G418 for 7 days. To confirm genomic integration, genomic DNA was extracted from stable cells using PureLink Genomic DNA Mini Kit (Invitrogen) and used as a template to amplify the integrated cDNA in the AAVS1 locus using primers from Appendix A. Complementation was then confirmed by Sanger sequencing.

### 2.9. Olaparib and Talazoparib Sensitivity Assays

PARP inhibitors, olaparib and talazoparib, sensitivity assay and cell imaging were performed as described previously [61]. Cells were treated for four days with concentrations ranging from 0 (DMSO) to 2.5 μM of olaparib or 0 (DMSO) to 40 nM of talazoparib. Cell viability was expressed as percentage of survival in olaparib-treated or talazoparib-treated cells relative to vehicle (DMSO)-treated cells. Results represent the mean ± standard error of the mean (SEM) of at least three independent experiments, each performed in triplicate.

### 2.10. Immunofluorescence Analysis

U2OS RAD51D cells stably complemented with either RAD51D WT or the p.Ser46Cys were seeded into Corning 96-Well Half Area High Content Imaging Film Bottom Microplate at 7000 cells per well. Then, 18h later, cells were irradiated with 5 Gray and processed for immunofluorescence 4 h post-irradiation. Unless otherwise stated, all immunofluorescence dilutions were prepared in PBS and incubations performed at room temperature with intervening washes in PBS. Cell fixation was carried out by incubation with 4% paraformaldehyde for 10 min followed by 100% ice-cold methanol for 5 min at −20 °C. Cells were permeabilised in 0.2% Triton X-100 for 5 min followed by a quenching step using 0.1% sodium borohydride for 5 min. After blocking for 1 h in a solution containing 10% goat serum and 1% BSA, cells were incubated for 1 h with primary antibody anti-RAD51 (1:5000, Bioacademia #70-001) or anti-phospho-Histone H2A.X (Ser139) (1:5000, Millipore, #05-636), combined with anti-Geminin (1:7000, Abcam #ab104306 or Proteintech #10802-1-AP) all diluted in 1% BSA. Secondary antibody labelling used Alexa Fluor 488 goat anti-rabbit (Invitrogen, #A-11008) and Alexa Fluor 647 goat anti-mouse (Invitrogen, #A21235) or Alexa Fluor 488 goat anti-mouse (Invitrogen, #A-11001) and Alexa Fluor 647 goat anti-rabbit (Invitrogen, #A21244), diluted at 1:1000 in 1% BSA for 1 h. Nuclei were stained for 10 min with 1 mg/mL 4,6-diamidino-2-phenylindole (DAPI). Z-stack images were acquired on a ZEISS Celldiscoverer 7 automated microscope using a 50× water immersion objective and analysed for RAD51 or gH2AX foci formation with ZEN Blue software 3.2 (ZEISS). Data from three independent trials were analysed for outliers using the ROUT method (*Q* = 1.0%) in GraphPad Prism v8.0 and the remaining were reported in a violin plot.

### 2.11. Protein Expression and Immunoblotting Analyses of RAD51D

Total soluble protein extraction and immunoblotting were performed as previously described [62]. For RAD51D detection, a polyclonal antibody (#ab202063, Abcam, US) was used at a 1:1000 dilution. Mouse monoclonal anti-vinculin (#V9131, Sigma, US) at 1:200,000 dilution was used as the loading control. Horseradish peroxidase-conjugated anti-rabbit IgG or anti-mouse at 1:10,000 dilution (Jackson Immuno Research, US) was used as secondary antibodies.

## 3. Results

### 3.1. Identification and Characteristics of Candidate Variants

By extracting all variants located in protein encoding and splice-site regions of *RAD51C* and *RAD51D* in WES data, we identified a total of 8 variants in 20 familial cases and 16 variants in 53 sporadic cases of OC (Figure 1 and Appendix A). From this list, we identified five candidate variants in these genes that fulfilled our selection criteria (Figure 1B): two missense variants in *RAD51C* c.414G>C; p.Leu138Phe and c.705G>T; p.Lys235Asn, two missense variants in *RAD51D* c.137C>G; p.Ser46Cys and c.620C>T; p.Ser207Leu and a nonsense variant *RAD51D* c.694C>T; p.Arg232Ter.

The variants were identified in nine OC cases: 17.6% (3/17) of OC families and 11.3% (6/53) of sporadic early-onset cases (Table 1). Our results include the identification of newly reported variants in FCs: two carriers of *RAD51C* c.414G>C; p.Leu138Phe, one of *RAD51C* c.705G>T; p.Lys235Asn, one of *RAD51D* c.137C>G; p.Ser46Cys and one of *RAD51D* c.694C>T; p.Arg232Ter. In addition, our results include four carriers of *RAD51D* c.620C>T; p.Ser207Leu, a variant previously reported to occur in more than one FC OC case by our group [28].

Our candidate variants are rare in various cancer-free populations of non-FC and European ancestry based on surveying available genetic data (Table 1 and Appendix A). All candidate variants have a MAF ≤ 0.0001 in the non-cancer population represented in the gnomAD database and in cancer-free women in the FLOSSIES database.

All variants were predicted to affect highly conserved loci and the four missense variants to be damaging by at least one of our in silico tools that we selected based on their best performance (Table 1). Four different in silico tools also predicted that the *RAD51C* c.705G>T; p.Lys235Asn located at the 5′ splice-donor site would affect transcript splicing.

The ClinVar database and/or ACMG guidelines classified *RAD51C* c.414G>C; p.Leu138Phe and c.705G>T; p.Lys235Asn and *RAD51D* c.694C>T; p.Arg232Ter as likely pathogenic or pathogenic. In contrast, there were conflicting classifications reported for *RAD51D* c.137C>G; p.Ser46Cys, such as VUS and likely benign in the ClinVar database and VUS by ACMG guidelines and for *RAD51D* c.620C>T; p.Ser207Leu, such as pathogenic/likely pathogenic in the ClinVar database and VUS by ACMG guidelines.

None of the five candidate variants were found to co-occur in the nine carriers of these variants. We also reviewed WES data of these nine carriers for the presence of pathogenic variants in any of the known risk genes for OC based on the National Comprehensive Cancer Network (NCCN) for clinical practice in oncology guidelines (version 2021–Genetic/Familial High-Risk Assessment: Breast, Ovarian and Pancreatic): *BRCA1*, *BRCA2*, *MLH1*, *MSH2*, *MSH6*, *PMS2*, *BRIP1*, *RAD51C*, *RAD51D*, *PALB2*, *ATM* or *STK11*. All carriers were found not to carry a pathogenic or likely pathogenic variant in any of these genes based on ClinVar or ACMG guidelines, with the exception of the *RAD51D* c.694C>T; p.Arg232Ter carrier who also harboured a pathogenic variant in *BRCA1* c.1462dupA; p.Thr488AsnfsTer2. Interestingly, this known *BRCA1* pathogenic variant has not been previously reported in the FC populations.

### 3.2. Carrier Frequency of Candidate Variants in OC Cases and Cancer-Free Controls of FC Ancestry

We compared the carrier frequencies of our candidate variants in different FC groups comprised of cancer cases, regardless of *BRCA1* or *BRCA2* pathogenic variant status, and cancer-free controls (Table 2). Pair-wise comparisons of carrier frequencies were performed using data from each cancer group and sequencing-based controls (see Appendix A). The highest overall carrier frequency was among carriers of *RAD51D* c.620C>T found in the sporadic group. Frequencies of this variant ranged from 2.3% (1/44) in OC families having at least two OC cases to 3.4% (15/438) in sporadic cases. Notably, all carriers were among the pathogenic *BRCA1* or *BRCA2* variant-negative cases as determined by previous studies which included whole gene or targeted genotyping of FC variants in these study groups (see Appendix A). The carriers in the sporadic cases included the previously identified carriers of this variant (3.8%; 13/341) [28]. In contrast, the carrier frequencies of each of the other variants were lower in the cancer groups. The carrier frequencies of these variants ranged from 0% to 2.3% (1/44) in OC families harbouring *RAD51C* c.414G>C, where the carrier was from a pathogenic *BRCA1* or *BRCA2* variant-negative family, and from 0% to 0.2% (1/438) for those harbouring *RAD51C* c.705G>T, *RAD51D* c.137C>G or c.694C>T variants. None of the index OC or BC cases from 56 HBOC families were found to harbour any of our candidate variants. Our targeted genotyping assays or review of available WES data revealed that none of the carriers identified in the cancer study groups (Table 2) also carried another one of our *RAD51C* or *RAD51D* candidate variants.

It was not possible to perform pair-wise comparisons to further assess carrier frequency of our variants in cancer study groups and genotyping-based controls. No carriers of *RAD51C* c.414G>C or *RAD51D* c.620C>T were identified in the genotyping-based data of 8493 population-matched controls. Carriers of *RAD51C* c.705G>T, *RAD51D* c.137C>G or *RAD51D* c.694C>T could not be identified in the same data and were not available in the HRC.r1 haplotype reference panel used for imputation from SNP array data.

Although the cancer study groups were independently derived for previous research purposes [21,23,24,34,63], we cannot exclude the possibility that individuals were recruited to multiple study groups. Based on the unique RRCancer biobank sample reference number, we are only aware of nine samples where the same case was included in two different study groups (Appendix A). We therefore performed a permutation analysis with 5000 random allocations of the observed variants to the participants across the three cancer groups (44 OC families, 56 HBOC families and 438 sporadic OC cases) and the two control groups (1025 sequencing-based and 8493 genotyping-based controls) investigated this study. When compared to sequencing-based controls, the permutation analysis presented evidence for a higher variant rate among all cases (*p* = 0.026), OC families (*p* = 0.015) and all families (*p* = 0.026). When examining only *RAD51D* c.620C>T, which was captured in data from both sequencing-based and genotyping-based controls, permutation analysis provided evidence for a higher frequency of this variant in all cancer cases versus all controls (*p* = 0.0098) and in all familial cases versus all controls (*p* = 0.014). The permutation analysis also allowed us to estimate the family-wise error rate for all tests performed: in 6.7% of the permutations, we found that at least one of the five tests demonstrated significance at *p* < 0.05, reflecting a type-1 error rate of potential concern. However, our permutation analysis also demonstrated that it was highly unlikely for all five comparisons to result in a naïve *p*-value < 0.05 simultaneously (permutation study *p* = 0.0002).

### 3.3. Clinico-Pathological Characteristics of OC Variant Carriers

The histopathological and clinical characteristics available for the 6 *RAD51C* and 28 *RAD51D* variant carriers are shown in Appendix A, which also includes known personal history of cancer. The pedigrees of selected carriers are shown in Appendix A, anonymized to only show information relevant to this study to protect the identity of participants. Thirteen of 28 *RAD51D* c.620C>T carriers from a previous study were also included for comparative purposes as their associated clinical features had not been reported [28]. Features of carriers of *RAD51C* c.705G>T (*n* = 2) and *RAD51D* c.137C>G (*n* = 1) and c.620C>T (*n* = 7) that were identified by targeted screening of an additional 538 cases of OC of FC ancestry were also included in Appendix A.

Most OC carriers of our candidate variants had HGSC (31/34), which is the most common subtype of epithelial OC [64]. Three other carriers had either a high-grade endometrioid adenocarcinoma, serous carcinoma of unknown grade or OC of undisclosed histopathology. A query of the OCAC data, which only revealed summary statistics for one of our candidates, showed statistical differences in the frequency of *RAD51D* c.620C>T carriers having HGSC (OR_Log2_ = 17.2; *p* = 0.00001) versus controls (Appendix A). This observation is consistent with our query of *BRCA1* c.4327C>T in OCAC data (OR_Log2_ = 1.211; *p* = 0.009051), the most common pathogenic FC OC risk allele as a comparator, where we found statistical differences in the frequency of carriers of this *BRCA1* variant having the high-grade serous subtype OC versus controls.

The average age of OC diagnosis in carriers of 58.5 years (median 59 years [age range = 42–77; SD ± 9.0 years]) was comparable to the average age of OC diagnosis in the general population being 60 years of age [28]. Fifty-three percent (18/34) of carriers were diagnosed before the age of 60 years, where 21% (7/34) were diagnosed before the age of 50 years.

Given the high frequency of *RAD51D* c.620C>T carriers in our OC cases, it was possible to compare clinical data of carriers (15/438) of this *RAD51D* variant with carriers (15/438) [21,28] of a frequently occurring variant in *BRCA1* c.4327C>T [21,23,24,34,65], previously reported in our investigation of the same sporadic OC study group [34]. The average and median ages at diagnosis of *RAD51D* variant carriers was approximately 59 years (age range = 46–74; SD ± 8.4 years). This was older than the average and median ages of diagnosis of 54 years observed in *BRCA1* variant carriers (age range = 36–76; SD ± 11 years (*p* = 0.15; 95% CI: −1.96 to 12.49; two tailed *p* value, unpaired *t*-test). The average survival among the 15 *RAD51D* c.620C>T carriers was 81.9 months (median 69 months [range = 1–195 months]) which was longer than the average survival of 67.1 months among *BRCA1* c.4327C>T carriers, though not significantly different (median 52 months [range = 10–168 months]; *p* = 0.46; 95% CI: −25.86 to 55.33, two-tailed *p* value, unpaired *t*-test). There were 46.7% (7/15) of *RAD51D* c.620C>T carriers and 60% (9/15) of *BRCA1* c.4327C>T carriers who had died of OC.

### 3.4. LOH Analyses of RAD51C and RAD51D Loci in OC Tumour DNA from Candidate Variant Carriers

Evidence of partial or complete loss of the wild-type allele was observed in tumour DNA from at least one carrier of each type of missense candidate variant (Table 3) as tumour DNA was not available for all variant carriers. Although our assays were not performed in DNA samples extracted from sections enriched for cancer cells, in five cases harbouring *RAD51D* c.137C>G or c.620C>T (PT0058, PT0071, PT0075, PT0076 and PT0077), there was clear evidence of loss wild-type allele in the analysis of tumour DNA extracted from FFPE. These findings suggest that partial loss or allelic imbalance observed with some samples may be an indication of contaminating normal stromal cells, although OC tumour specimens are often abundant in cancer cells [66]. Interestingly, only the *RAD51D* c.620C>T germline allele was detectable in both OC tumours from a bilateral OC case by Sanger sequencing of tumour DNA (Figure 2). This observation suggests the possibility that somatic loss of the wild-type allele preceded clonal expansion in the tumourigenesis of OC in this carrier.

### 3.5. In Vitro Investigation of Aberrant Splicing of RAD51C c.705G>T

We established *RAD51C* c.705G>T carrier- and non-carrier-derived LCLs and performed RT-PCR on the extracted RNA to determine if the genomic position of the variant affected splicing consistent with predictions based on the application of our selected in silico tools (Table 1). RT-PCR analyses showed two different size bands from the c.705G>T carrier-derived LCLs but not in controls (Figure 3A), suggesting that this variant affected splicing of the transcript. Although we could not verify the presence of exon 4 in the non-carrier as cDNA no longer available, Sanger sequencing verified the absence of the entire exon four in the variant carrier-derived cDNA (Figure 3B,C), suggesting that exon skipping had occurred.

### 3.6. In Cellulo Investigation of RAD51D p.Ser46Cys

We performed in cellulo assays of RAD51D p.Ser46Cys due to conflicting reports of its clinical significance and paucity of data concerning its effect on biological function. We selected this candidate variant for further study as the biological impact of our other remaining missense candidate variants, RAD51C p.Leu138Phe [5] and RAD51D p.Ser207Leu [28] have been reported, or have been inferred as biologically relevant as described above for the effects of splicing on *RAD51C* c.705G>T.

Given the role of RAD51D in HR function, we investigated sensitivity to the PARP inhibitor, olaparib, taking advantage of the synthetic lethal interaction between loss of HR function and PARP inhibition [62]. Using HeLa cells, RAD51D-knock-down cells were more sensitive to olaparib (Appendix A). Complementation with RAD51D WT siRNA resistant construct restored sensitivity to endogenous levels, while RAD51D p.Ser46Cys siRNA resistant construct failed to rescue the viability of RAD51D knock-down cells, showing olaparib sensitivity similar to cells complemented with the empty vector.

Immunoblotting 24 h post-transfection showed that RAD51D p.Ser46Cys protein was weakly expressed compared to WT (Appendix A). As the expression of RAD51D p.Ser46Cys was lower than the WT, we then investigated if the reduced expression of the variant protein could be due to protein instability by examining the protein’s half-life in RAD51D knock-down cells transfected with either FLAG-RAD51D WT or p.Se46Cys. Cells were then exposed to cycloheximide (CHX) to inhibit protein synthesis and pellets for protein extraction were collected at the indicated time points. Over the time course RAD51D protein levels were reduced for both isoforms, however, the effect was more pronounced in cells expressing the p.Ser46Cys variant when compared to the WT (Appendix A). RAD51D WT protein starts to reduce at after six hours, while the RAD51D p.Ser46Cys protein is hardly visible at four hours (Appendix A). We were able to recapitulate our findings in an OC cell line background (Appendix A). In OVCAR-3 cells, the p.Ser46Cys variant protein is also expressed at a reduced level. These observations suggest that RAD51D p.Ser46Cys is unstable and affects cellular sensitivity to olaparib.

To further investigate the functionality of the p.Ser46Cys variant and eliminate the effect of endogenous RAD51D, we used U2OS RAD51D KO cells stably complemented with either the WT or the p.Ser46Cys variant using the AAVS1 genomic editing system (Figure 4A) [59,60]. Although RAD51D KO cells were successfully complemented with either WT and p.Ser46Cys, as confirmed by Sanger sequencing, the p.Ser46Cys variant was also weakly expressed in this cell line when compared to the WT (Figure 4B,C).This is in agreement with the observation that U2OS RAD51D cells harbouring the p.Ser46Cys variant failed to complement survival when exposed to olaparib and talazoparib (Figure 4D,E).

RAD51D is required to facilitate RAD51 filament formation and proper repair of damage-induced double-strand breaks [67]. Therefore, we evaluated both RAD51 and γH2AX foci formation after treatment with 5 Gray of ionizing radiation in S/G2-cells (Appendix A). As expected, a decrease in the mean number of RAD51 foci per cell was observed in RAD51D deficient cells and expression of RAD51D WT partially rescued this phenotype, while rescue was less obvious in cells expressing the p.Ser46Cys (Appendix A). Moreover, after ionizing radiation, the p.Ser46Cys variant also exhibited elevated levels of γH2AX foci, while RAD51D WT cells were able to rescue the increased γH2AX foci formation observed in the RAD51D deficient cells (Appendix A). Altogether, these results indicate an impaired HR functionality leading to increased DNA double-strand breaks in cells expressing the RAD51D p.Ser46Cys variant.

## 4. Discussion

Our WES analysis of 73 familial and sporadic early-onset OC FCs of Quebec identified five candidate variants in *RAD51C* and *RAD51D*. The genetic analyses of additional FC OC study groups confirmed that *RAD51D* c.620C>T, previously reported in sporadic OC cases by our group [28], occurs at a high frequency in FCs with this disease. This observation is likely due to the unique genetic architecture of FCs of Quebec that has been attributed to common founders of this population [20,21,22]. OC cases harbouring other variants were found once for the nonsense *RAD51D* c.694C>T and twice for the missense *RAD51D* c.137C>G or *RAD51C* c.414G>C, suggesting that FCs are more genetically heterogenous population than other populations with common founders where few frequently occurring variants have been reported [21]. We identified a total of four OC cases harbouring *RAD51C* c.705G>T suggesting that these individuals might also share common ancestors in FCs. Indeed, another OC case harbouring this *RAD51C* variant was identified in a woman diagnosed with a HGSC of unknown origin (likely upper genital tract) in a hereditary cancer clinic by medical genetic panel testing and was provided to us at the conclusion of this study (Appendix A).

The differing carrier frequencies of our variants likely reflect genetic drift due to the waves of localized expansion of the FC population that occurred in Quebec since the founding of this European population in 1608 [21,27]. This change in genetic architecture of the FC population over time has been proposed to account for the varying frequencies of carriers of different pathogenic variants in *BRCA1* and *BRCA2* in this population [20,21,22]. We cannot exclude the possibility that those that harbour the same variant are closely related as familial associations were not available nor known for all OC cases investigated in this study. We have not determined identity by descent of the more frequently occurring variants as for *RAD51D* c.620C>T [28], due to paucity of cases harbouring such variants in our investigation. It is plausible given the history of the FC population in Quebec that individuals harbouring these variants share common ancestors as we have shown in our studies of the most frequently occurring pathogenic variants in *BRCA1*, *BRCA2*, and *PALB2* [21,23,24,25,26,63,68,69].

All variants except *RAD51C* c.705G>T have been reported in OC cases from other populations in the published literature. Our literature review of OC or BC cases with pedigrees (Appendix A) showed that those harbouring our candidate variants had a family history of OC (Appendix A). It has been shown that pathogenic *BRCA1* or *BRCA2* variant-negative families with a family history of at least two OC cases are more likely to harbour a pathogenic variant in *RAD51C* or *RAD51D* though the overall carrier frequency is lower than that found for pathogenic *BRCA1* or *BRCA2* variant-carriers [5,6,7,17,70]. Our findings showed that the average ages at diagnosis of all cases harbouring *RAD51C* and *RAD51D* variants are 58.0 and 58.6 years, respectively, which is comparable with a recent population-based study [71]. These observations are consistent with age at diagnosis of cases harbouring pathogenic variants as reported in the NCCN guidelines [72]. However, our study showed that 21% (7/34) of women harbouring our candidate variants developed OC before the age of 50 years, where the youngest was diagnosed at age of 42. We also showed a higher frequency of the sporadic early-onset OC cases harbouring *RAD51D* variants which is consistent with a previous report [73]. Although we are not able to estimate risk, our data suggests that penetrance might vary in those harbouring pathogenic variants in *RAD51C* or *RAD51D*.

Candidate variants were prioritized for genetic analyses in our study groups based on results from high performance in silico tools for missense variants [41,43]. RAD51D p.Arg232Ter is predicted to affect RAD51D protein production due to premature amino acid termination eliciting nonsense-mediated mRNA decay, rendering this LoF variant compatible with conferring risk for OC [70,74,75] and its classification as pathogenic. The aberrant function of the RAD51C p.Leu138Phe and RAD51D p.Ser207Leu protein variants have been reported independently, where there was a significant reduction of RAD51 foci affecting HR function in the complemented cell lines [5,28]. RAD51D p.Ser207Leu has been shown to disrupt the direct interaction of RAD51D and XRCC2 in RAD51B-RAD51C-RAD51D-XRCC2 (BCDX2) complex reducing HR function [28,76] rendering cells sensitive to PARP inhibitor, olaparib [28] in RAD51D KO cell lines [6]. Our investigation clearly demonstrates that cancer cell lines, including an epithelial ovarian adenocarcinoma cell line characteristic of HGSC OC disease, complemented with the RAD51D p.Ser46Cys have impaired protein expression. Two of the cell lines tested are also sensitive to olaparib (see Figure 4 and Appendix A). Sensitivity may be explained by the weak expression of the protein variant which would impact HR function as was observed by the reduced RAD51 foci formation and increased in γH2AX foci. However, further assays are required to elucidate the underlying mechanism of HR deficiency that resulted in olaparib and talazoparib sensitivity. *RAD51C* c.705G>T; p.Lys235Asn is an interesting missense variant as the nucleotide alteration occurs at the 5′ splice-donor site of the coding region which is predicted to affect splicing by skipping exon four, as was demonstrated by our assays of cDNA from carrier-derived LCLs. We were unable to confirm the presence of exon 4 in the cDNA from non-carrier derived LCLs as cDNA was no longer available. However, our results are consistent with a recent report, which was published during the course of this study, showing that the entirety of exon four of *RAD51C* was excluded from the transcript using a splicing reporter minigene system of this variant [77]. Interestingly, this report showed compelling evidence that only the transcript lacking exon 4 was only transcribed. Although we cannot exclude the possibility that the exon skipping occurs 100% of the time in cell line model systems applied to assay transcription of this variant, it is difficult to demonstrate that exon skipping also occurs in progenitor cells biologically relevant to the development of OC in carriers. Future studies investigating such variants in cancer predisposition models might be helpful. Notable is that exon 4 of RAD51C encodes the Walker-B ATPase motif (see Figure 3C) that is critical for RAD51C function in the HR pathway [78]. Although the mechanisms of aberrant *RAD51C* and *RAD51D* in conferring risk to OC is unknown, our LOH analyses of tumour DNA from carriers are consistent with independent studies that have demonstrated loss of the protein function in tumour cells. Moreover, our LOH analyses of one of the *RAD51D* c.620C>T carriers with bilateral disease suggests that the loss of wild-type *RAD51D* allele was an early event in ovarian tumourigenesis. Collectively, our findings support the application of our bioinformatic pipeline of WES data and selected predictive tools to identify candidate missense variants in *RAD51C* and *RAD51D* suitable for functional validation.

## 5. Conclusions

We were able to identify *RAD51C* and *RAD51D* candidate variants implicated in familial and sporadic OC using our strategy of investigating the germline DNA of the genetically unique FC population that may also be relevant to non-FC populations. Our filtering and prioritizing criteria allowed us to focus on the role of missense variants as candidate OC risk alleles, variants that are more difficult to assess using genetic strategies due to inferences in their role in abrogating gene function. To the best of our knowledge, this is the first report describing *RAD51C* c.705G>T; p.Lys235Asn in the context of hereditary OC, and purporting the clinical relevance of *RAD51D* p.Ser46Cys by our in cellulo assays including olaparib sensitivity. Collectively, our findings suggest that our variants are all likely pathogenic, further supporting the role of *RAD51C* and *RAD51D* in conferring risk to OC.

## Figures and Tables

**Figure 1 cancers-14-02251-f001:**
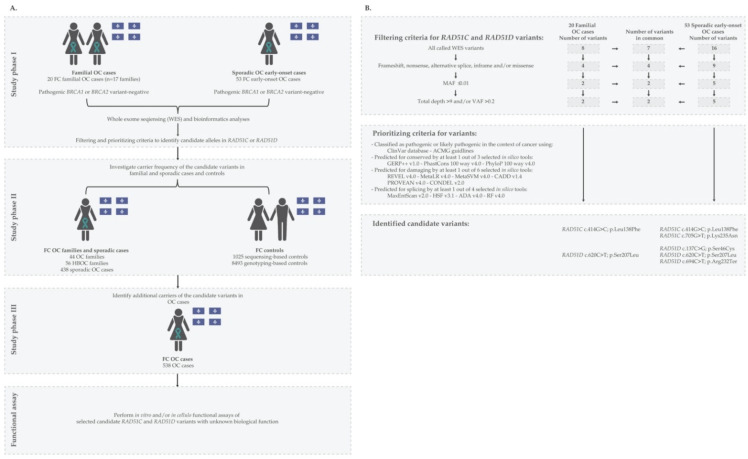
Study design and criteria for identifying candidate variants in *RAD51C* and *RAD51D*. (**A**) Schematic diagram summarizing phase I of the study for identifying candidate variants in ovarian cancer (OC) cases, phase II of the study for determining the carrier frequency of the candidate variants in familial and sporadic OC cases, Hereditary Breast and Ovarian Cancer (HBOC) syndrome families and controls from the French Canadian (FC) population of Quebec, and phase III of the study for identifying additional carriers; teal ribbon signifies women with OC; and diagrams contain provincial flag of Quebec denoting geographic ascertainment of cases and controls; and (**B**) Schematic diagram presenting the filtering and prioritizing criteria applied to identify candidate variants (see Appendix A). Variants were prioritized using different in silico tools for conservation: Genomic Evolutionary Rate Profiling v1.0 (GERP++ v1.0 (score ≥ 2.0)); Phylogenetic p value v4.0 of 100 vertebrates (PhyloP 100 way v4.0 (score ≥ 0.2)) and PHAST Conservation v4.0 of 100 vertebrates (PhastCons 100 way v4.0 (score ≥ 0.4)); for predicting damaging effects based on their best predictive performance: Rare Exome Variant Ensemble Learner v4.0 (REVEL v4.0 (score ≥ 0.4)); Meta-analytic Logistic Regression v4.0 (MetaLR v4.0 (score ≥ 0.4)); Meta-analytic support Vector Machine v4.0 (MetaSVM v4.0 (score ≥ 0.0)); Combined Annotation Dependent Depletion v1.3 (CADD v1.4 (Phred score ≥ 20)); Protein Variation Effect Analyzer v4.0 (PROVEAN v4.0 (score ≥ −2.5)); and CONsensus DELeteriousness v2.0 (CONDEL v2.0 (score ≥ 0.4)); and for affecting alternative splicing: Maximum Entropy Estimates of Splice junction strengths v2.0 (MaxEntScan v2.0); Human Splicing Finder v3.1 (HSF v3.1); and two different Database Splicing Consensus Single Nucleotide Variant (dbscSNV) in silico tools: AdaBoost v4.0 (ADA v4.0 (score ≥ 0.4)) and Random Forest v4.0 (RF v4.0 (score ≥ 0.4)) (see Materials and Methods Section 2.2.).

**Figure 2 cancers-14-02251-f002:**
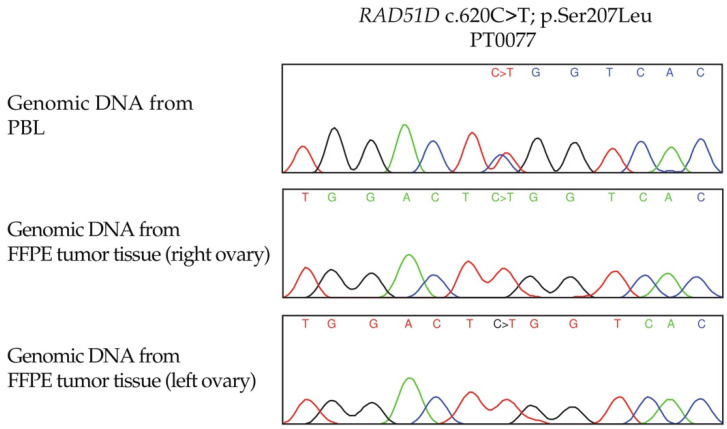
Loss of heterozygosity (LOH) analysis of a *RAD51D* c.620C>T carrier. Sanger sequencing chromatograms showing evidence of a complete loss of the wild-type variant in genomic DNA from formalin-fixed paraffin embedded (FFPE) tumour tissues from both ovaries.

**Figure 3 cancers-14-02251-f003:**
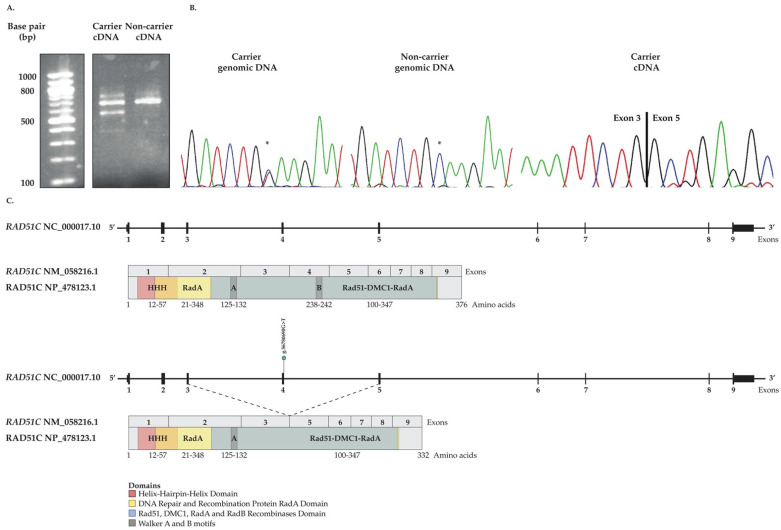
*RAD51C* c.705G>T effect on transcript splicing in carrier- and non-carrier-derived lymphoblastoid cell lines (LCLs) (see Appendix A). (**A**) Agarose gel of cDNA analysis showing different sized bands from carrier- compared to non-carrier-derived LCLs; (**B**) Sanger sequencing chromatograms of genomic and cDNA from carrier-derived LCLs, confirming the variant status as indicated with * in genomic DNA and showing skipping of exon four in cDNA; and (**C**) The upper panel depicting the wild-type *RAD51C* at the genomic, mRNA and protein level whereas the lower panel depicting the predicted effect of *RAD51C* c.705G>T at mRNA and protein level, resulting in skipping of exon four (44 amino acids) annotated according to genomic (NC_000017.10), mRNA (NM_058216.1) and protein (NP_478123.1) NCBI Reference Sequence (RefSeq) Database (ncbi.nlm.nih.gov/refseq/, accessed on 1 October 2021).

**Figure 4 cancers-14-02251-f004:**
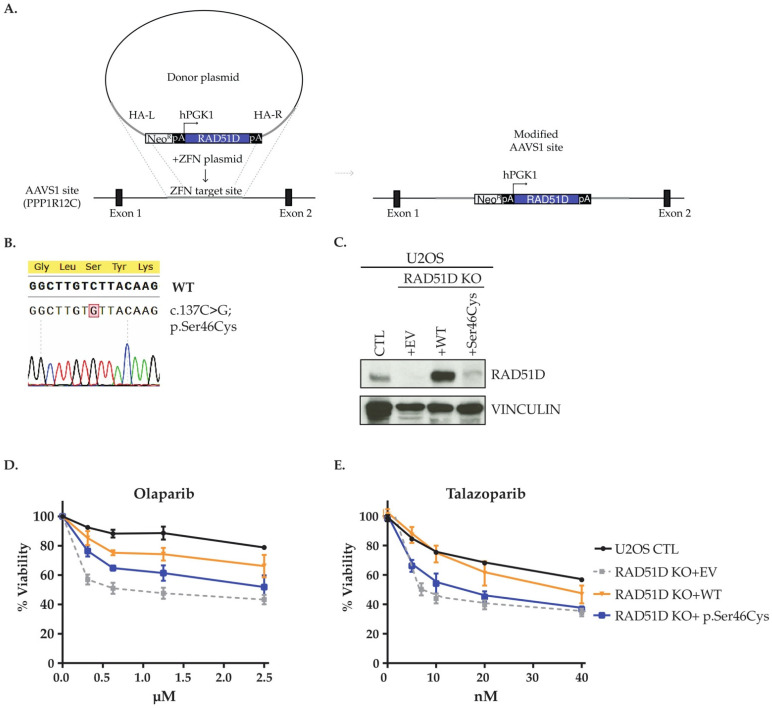
The RAD51D p.Ser46Cys variant impairs protein stability and function in U2OS RAD51D knock-out (KO) cells. (**A**) Scheme representing the AAVS1 genomic integration system used to complement the RAD51D KO U2OS cell line. (**B**) U2OS RAD51D KO cells complemented with the AAVS1 system were confirmed by Sanger sequencing. (**C**) Western blots of U2OS RAD51D KO cells stably complemented with wild-type (WT) or the p.Ser46Cys variant (see Appendix A); CTL was used as non-edited control and Vinculin was used as a loading control. (**D**,**E**) Survival curves of U2OS RAD51D KO cells stably complemented with the WT RAD51D, the RAD51D p.Ser46Cys variant or empty vector (EV) and plated in triplicate in a 96 well plate. Cell viability was monitored following (**D**) olaparib or (**E**) talazoparib treatments for 96 h and was assessed by counting remaining nuclei. Experiments were performed in three biological replicates.

**Table 1 cancers-14-02251-t001:** Characteristics of candidate variants identified in *RAD51C* and *RAD51D*.

Gene	*RAD51C*	*RAD51C*	*RAD51D*	*RAD51D*	*RAD51D*
**Genomic features (GRCh37/hg19)**					
RefSeq transcript no.	NM_058216.3	NM_058216.3	NM_002878	NM_002878	NM_002878
Genome change	g.56774063G>C	g.56780690G>T	g.33446137G>C	g.33430520G>A	g.33430317G>A
Coding change	c.414G>C	c.705G>T	c.137C>G	c.620C>T	c.694C>T
Protein change	p.Leu138Phe	p.Lys235Asn	p.Ser46Cys	p.Ser207Leu	p.Arg232Ter
**Number of carriers discovered (Phase I)**					
Familial OC cases (*n* = 20)	1	0	0	2	0
Sporadic OC early-onset cases (*n* = 53)	1	1	1	2	1
**Allele frequencies in gnomAD ^1^**					
Non-Finish European	0.00001 (1/102,736)	0.00001 (1/102,610)	0.0001 (16/118,138)	0.0001 (6/118,136)	0.00003 (4/126,578)
**Carrier frequencies in FLOSSIES ^2^**					
European	0 (0/7325)	0 (0/7325)	0.0002 (2/7325)	0.0003 (3/7325)	0.0001 (1/7325)
**Clinical classification ^3^**					
ClinVar (number of submissions)	Pathogenic/Likely pathogenic (7)	Conflicting (7): Likely pathogenic (1); Uncertain significance (6)	Conflicting (8): Uncertain significance (7); Likely benign (1)	Conflicting (11): Pathogenic (2); Likely pathogenic (6); Uncertain significance (3)	Pathogenic (15)
ACMG guidelines (classification codes)	Likely pathogenic (PS1; PM2; PP3; PP5)	Pathogenic (PS3; PM2)	Uncertain significance (PM2; PP3)	Uncertain significance (PS3; M2; PP3)	Pathogenic (PVS1; PM2; PP3; PP5)
**Predictions by in silico tools ^4^**					
GERP++ v1.0	Conserved	Conserved	Conserved	Conserved	Conserved
PhyloP 100 way v4.0	Conserved	Conserved	Conserved	Conserved	Conserved
PhastCons 100 way v4.0	Conserved	Conserved	Conserved	Conserved	Conserved
REVEL v4.0	Pathogenic	Benign	Pathogenic	Pathogenic	-
MetaLR v4.0	Tolerated	Tolerated	Tolerated	Damaging	-
MetaSVM v4.0	Tolerated	Tolerated	Tolerated	Damaging	-
CONDEL v2.0	Damaging	Tolerated	Damaging	Damaging	-
PROVEAN v4.0	Damaging	Tolerated	Damaging	Damaging	-
CADD v1.4	Damaging	Damaging	Damaging	Damaging	Damaging
ADA v1.1	-	Affecting splicing	-	-	-
RF v1.1	-	Affecting splicing	-	-	-
HSF v3.1	-	Affecting splicing	-	-	-
MaxEntScan v2.0	-	Affecting splicing	-	-	-

^1^ Allele frequencies in non-cancer controls from gnomAD v2.1.1 database (gnomad.broadinstitute.org). Allele frequencies in non-cancer controls from different populations from gnomAD v2.1.1 database are presented in Appendix A. ^2^ Carrier frequencies from non-cancer controls from Fabulous Ladies Over Seventy (FLOSSIES) database (whi.color.com/about) (see Appendix A). ^3^ Clinical classifications from ClinVar (ncbi.nlm.nih.gov/clinvar/) [38,39] and American College of Medical Genetics and Genomics (ACMG) guidelines and associated codes [40] based on last revision reviewed in October 2021 as PS1: Pathogenic Strong Level 1; PS3; Pathogenic Strong Level 3; PM2: Pathogenic Moderate Level 2; PP3: Pathogenic Supporting Level 3; PP5: Pathogenic Supporting Level 5; and PVS1: Pathogenic Very Strong Level 1. ^4^ Details of in silico tools applied: ADA v1.1: AdaBoost v1.1; CADD v1.4: Combined Annotation Dependent Depletion v1.4; CONDEL v2.0: CONsensus DELeteriousness v2.0; GERP++ v1.0: Genomic Evolutionary Rate Profiling v1.0; HSF v3.1; Human Splicing Finder v3.1; MaxEntScan v2.0: Maximum Entropy Estimates of Splice Junction Strengths v2.0; MetaLR v4.0: Meta-analytic Logistic Regression v4.0; MetaSVM v4.0: Meta-analytic Support Vector Machine v4.0; PhyloP 100 way v4.0: phylogenetic *p* value v4.0 of 100 vertebrates; PhastCons 100 way v4.0: PHAST Conservation v4.0 of 100 vertebrates; PROVEAN v4.0: Protein Variation Effect Analyzer v4.0; RF v1.1: Random Forest v1.1; REVEL v4.0: Rare Exome Variant Ensemble Learner v4.0. OC: ovarian cancer; RefSeq: reference sequence; and (-): Not applicable/available.

**Table 2 cancers-14-02251-t002:** Carrier frequency of candidate variants in French Canadian study groups and comparison of cancer cases to controls.

Variant	Study Groups	Cancer Case Tested	Number of Participants or (Families) per Group	Number of Carriers (%)	*p* Value ^1^
***RAD51C* c.414G>C**	OC families	OC	49 (44)	1/44 (2.3)	0.081
	HBOC families	OC or BC	56 (56)	0	-
	Sporadic OC cases	OC	438	0	-
	Sequencing-based controls	-	1025	1/1025 (0.1)	-
***RAD51C* c.705G>T**	OC families	OC	49 (44)	0	-
	HBOC families	OC or BC	56 (56)	0	-
	Sporadic OC cases	OC	438	1/438 (0.2)	0.299
	Sequencing-based controls	-	1025	0	-
***RAD51D* c.137C>G**	OC families	OC	49 (44)	0	-
	HBOC families	OC or BC	56 (56)	0	-
	Sporadic OC cases	OC	438	1/438 (0.2) ^2^	0.299
	Sequencing-based controls	-	1025	0	-
***RAD51D* c.694C>T**	OC families	OC	49 (44)	0	-
	HBOC families	OC or BC	56 (56)	0	-
	Sporadic OC cases	OC	438	1/438 (0.2) ^2^	0.299
	Sequencing-based controls	-	1025	0	-
***RAD51D* c.620C>T**	OC families	OC	49 (44)	1/44 (2.3)	0.081
	HBOC families	OC or BC	56 (56)	0	-
	Sporadic OC cases	OC	438	15/438 (3.4) ^3^	<0.0001
	Sequencing-based controls	-	1025	1/1025 (0.1)	-

^1^ Two-tailed *p* values calculated using Fisher’s exact test in pair-wise comparisons between carriers in cancer study groups and controls; not adjusted for multiple testing. ^2^ Carriers known to also have been part of the sporadic early-onset OC cases phase I study group (see Appendix A). ^3^ Thirteen of 15 *RAD51D* c.620C>T carriers were previously reported [28] (see Appendix A). BC: Breast cancer; HBOC: Hereditary breast and ovarian cancer syndrome; OC: Ovarian cancer; and (-): Not applicable.

**Table 3 cancers-14-02251-t003:** Loss of heterozygosity (LOH) analyses of tumour DNA from ovarian cancer carriers by Sanger sequencing.

Carrier ID ^1^	Gene	Coding Change ^2^	Protein Change	Germline Status	Laterality of Disease	LOH Analyses of Available DNA from Fresh Frozen Tumour			LOH Analyses of Available DNA from Formalin-Fixed Paraffin-Embedded Tumour	
						Right ovary	Left ovary	Laterality unknown or alternative tissue	Right ovary	Left ovary
PT0095	*RAD51C*	c.414G>C	p.Leu138Phe	Heterozygous	Unilateral (Left)	-	-	-	-	-
PT0094	*RAD51C*	c.414G>C	p.Leu138Phe	Heterozygous	Bilateral	-	-	Partial loss in ascites	-	-
PT0124	*RAD51C*	c.705G>T	p.Lys235Asn	Heterozygous	Bilateral	Partial loss	-	-	-	-
PT0125	*RAD51C*	c.705G>T	p.Lys235Asn	Heterozygous	Bilateral	-	Complete loss	-	-	-
PT0126 ^3^	*RAD51C*	c.705G>T	p.Lys235Asn	Heterozygous	Bilateral	Heterozygous	-	-	-	-
PT0127	*RAD51C*	c.705G>T	p.Lys235Asn	Heterozygous	Unknown	-	-	-	-	-
PT0143	*RAD51D*	c.694C>T	p.Arg232Ter	Heterozygous	Bilateral	-	-	-	-	-
PT0058	*RAD51D*	c.137C>G	p.Ser46Cys	Heterozygous	Bilateral	-	-	Heterozygous	Partial loss	-
PT0145	*RAD51D*	c.137C>G	p.Ser46Cys	Heterozygous	Bilateral	Partial loss	-	-	-	-
PT0080	*RAD51D*	c.620C>T	p.Ser207Leu	Heterozygous	Bilateral	-	-	Partial loss in omentum	-	-
PT0071	*RAD51D*	c.620C>T	p.Ser207Leu	Heterozygous	Bilateral	Partial loss	-	-	Partial loss	-
PT0073	*RAD51D*	c.620C>T	p.Ser207Leu	Heterozygous	Unilateral (Left)	-	-	-	-	-
PT0090	*RAD51D*	c.620C>T	p.Ser207Leu	Heterozygous	Bilateral	-	-	-	-	-
PT0078	*RAD51D*	c.620C>T	p.Ser207Leu	Heterozygous	Bilateral	-	-	-	-	-
PT0079	*RAD51D*	c.620C>T	p.Ser207Leu	Heterozygous	Unilateral (Left)	-	-	-	-	-
PT0089	*RAD51D*	c.620C>T	p.Ser207Leu	Heterozygous	Bilateral	-	-	-	-	-
PT0059	*RAD51D*	c.620C>T	p.Ser207Leu	Heterozygous	Bilateral	-	-	Complete loss in ovary	-	-
PT0065 ^3^	*RAD51D*	c.620C>T	p.Ser207Leu	Heterozygous	Bilateral	-	-	Heterozygous in ovary	-	-
PT0075 ^3^	*RAD51D*	c.620C>T	p.Ser207Leu	Heterozygous	Unilateral (Right)	Partial loss	-	-	Complete loss	-
PT0076	*RAD51D*	c.620C>T	p.Ser207Leu	Heterozygous	Bilateral	-	-	-	Complete loss	Partial loss
PT0077	*RAD51D*	c.620C>T	p.Ser207Leu	Heterozygous	Bilateral	-	-	-	Complete loss	Complete loss
PT0074 ^3^	*RAD51D*	c.620C>T	p.Ser207Leu	Heterozygous	Bilateral	Partial loss	-	-	-	-
PT0144	*RAD51D*	c.620C>T	p.Ser207Leu	Heterozygous	Bilateral	-	-	Heterozygous	-	-

^1^ The 13 carriers with high-grade serous ovarian carcinomas (HGSC) previously reported are not included in this table. ^2^ Transcripts of RAD51C (NM_058216.3) and RAD51D (NM_002878.4) are based on the NCBI Reference Sequence (RefSeq) database (ncbi.nlm.nih.gov/refseq/). ^3^ The DNA was extracted from tumour samples post-chemotherapy treatment. (-): Tumour DNA not available or failed analyses.

## Data Availability

WES data for familial and sporadic OC cases, CARTaGENE, MNI and Gen3G will be returned to their respective biobanks at the conclusion of our study of OC predisposing genes which is still ongoing. For more information concerning these data contact Patricia N. Tonin at patricia.tonin@mcgill.ca. The data from the analyses of investigation of OCAC, FLOSSIES and gnomAD are available from each of these data resource banks.

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
