# Peer review of "The Genetic and Molecular Analyses of RAD51C and RAD51D Identifies Rare Variants Implicated in Hereditary Ovarian Cancer from a Genetically Unique Population"

_cancers, 2022, doi:10.3390/cancers14092251_

Round 1

Reviewer 1 Report

I would like to thanks the authors for the considering my suggestions. I feel completely satisfied with the current version – I have no other comments. Congratulations for a very nice work. I do recommend the article in the current form for the publishing.

Author Response

Please see the attached cover letter with responses to reviewers.

Reviewer 2 Report

Authors did not fully address my concerns.

#1, PCR often has nonspecific bands. Without sanger sequencing or RNA sequencing, is hard to convince the readers that these bands are indeed what they expected are.

#2, unless authors probe with FLAG antibody, we are not sure the upper band is indeed the FLAG- RAD51D: it can be a nonspecific band; or the upper band is WT and the lower band is a degradation band. U2OS is an osteosarcoma cell line, not ovarian cancer. I don’t understand why the authors keep using different cell lines for the ovarian cancer study. In addition, success with PCR amplify of the inserted DNA at AAVS1 locus don’t mean the inserted fragment is successful transcripted/expressed: the inserted fragment can be quickly silenced by methylation. Again, authors need to show western blot instead of indirect evidence.

#3, please show us the detail of the siRNA resistance expression construct. Again, this is based on the assumption that the lower band is WT, but this assumption is not necessarily true and need to be supported by data.

#4, I totally disagree with the statement that “conventional chemiluminescence and autoradiography which could be misleading”. The figure still has the same issue of the protein half-life, remove the quantification image doesn’t affect this observation.

Minor points:

Authors should try another FLAG antibody. Usually FLAG antibody work quite well.

In the table of rebuttal letter, authors indicated that they used mM of Olaparib. This is super high.

Author Response

(The authors gave the same response as above.)

Reviewer 3 Report

the authors have sufficiently addressed my comments.

Author Response

(The authors gave the same response as above.)

Reviewer 4 Report

While my main concern remains that the novelty of the findings is limited, I acknowledge the authors' effort to improve the manuscript and adequately address the points raised by the reviewers.

Author Response

(The authors gave the same response as above.)

Round 2

Reviewer 2 Report

na

This manuscript is a resubmission of an earlier submission. The following is a list of the peer review reports and author responses from that submission.

Round 1

Reviewer 1 Report

This study takes a very deep dive into ovarian cancer in the French Canadian population and identifies variants in RAD51C/D and further characterizes it.  It is a large cohort, and conduct a gene discovery cohort through exome sequencing and validate these findings on larger datasets.

The authors further extend their findings to tumour analysis for LOH to determine if the germline RAD51 variants are driver mutations.  They reach clinical implications by studying the RAD51 variants in experiments examining PARPi sensitivity.

This study is sound, and provides meaningful insight for RAD51 variants.  Moderately penetrant variants in ovarian cancer are rare, and unlike BRCA1/2, large studies to validate pathogenicity are challenging.  This group does an excellent job in clarifying this.

I only have very minor comments on style.

  1. In the introduction, line 101, they state that PARPi, olaparib.  There are many PARPi why did they call out olaparib
  2. instead of carrying variants, the authors should use harbor (Jarvik PMID 27657676)
  3. The authors conclude that the variants found should now be called pathogenic despite previous conflicting reports.  A table outlining the rationale for this using the respective ACMG codes should be added to the main text, or even table 1.

Author Response

Response is included in the cover letter and attached.

Reviewer 2 Report

The presented draft of the article is a comprehensive analysis of predisposing factors influencing the risk of ovarian cancer. It is a classically constructed analysis using a unique set of cases and relevant controls.

I have no comments to the workflow and to the presentation of the results, with one exception – the functional analysis. The functional characterisation of missense variants is generally a bit tricky.

Given the used model system - which consists of a single cell line, moreover histologically unrelated to ovarian tissue; the use of siRNA knock-down; but mainly due to lack of controls, I have serious doubts about the relevance of the results. It would be more than appropriate to supplement the functional analysis of the missense variant with a set of positive and negative controls (and not “just” wt) – classified pathogenic (ideally both nonsense and missense), and benign variants that will clearly demonstrate that the design model system works and is able to distinguish between them.

I consider the draft article to be of high quality and beneficial. I recommend it for publication after supplementing the relevant missing results of the functional analysis.

Author Response

(The authors gave the same response as above.)

Reviewer 3 Report

Figure2a, authors should perform sanger seq of all the bands in the carrier cDNA lane. Right panel, Author should show the sanger result of non-carrier cDNA as well.

Figure 3b, based on the western blot result presented, it is hard to convince readers that authors have successfully transfected and expressed the mutant RAD51D in Hela cells (notably, Hela cell is not ovarian cancer, authors should use an ovaria cancer cell such as CAOV3, SKOV3 etc. or non-cancerous ovarian cell line). Thus, the phenotype presented in figure 3A is likely due to the consequence of silencing of RAD51D, instead of the Ser46Cys RAD51 mutant (in the Figure 2f of the ref27, which published previously by the same group of authors, they know how to do the experiment in an appropriate way). The authors then suggest the Ser46Cys mutation leads to a shorter protein hard of RAD51. However, the western blot result in Figure 3c shows the opposite result: the wild type RAD51 actually have a shorter protein half-life than the mutant (in time point 4 hours, the lane only has about 10-20% of RAD51 comparing with time point zero). The curves in figure 3d totally not match with the western blot result if the authors do the densitometry analysis of the bands.

Minor points:

In order to show endogenous and ectopic expressed RAD51D, Authors should probe the western blot with both RAD51 and Flag antibodies.

Figure 3a, authors should calculate the IC50 of Olaparib in each cell line. The range of the inhibitor concentration is still not reaching the 50% inhibition.

Author Response

(The authors gave the same response as above.)

Reviewer 4 Report

The manuscript by Alenezi et al., describes the analysis of the RAD51C and RAD51D genes in a series of 20 and 53 familial and sporadic ovarian cancer cases respectively. They find five candidate variants (four missense and one nonsense), all of them previously described, and some of them already classified as pathogenic or likely pathogenic based on previous functional assays. My main concern regarding this manuscript is that there is no novelty and very little adding to the field. RAD51C/D are well established ovarian cancer susceptibility genes and I don´t think that the study of a genetically unique population, does add much further information in this particular case. There is one variant that is presented as firstly described related to ovarian cancer, c.705G>T; Lys235Asn in RAD51C, however, the variant is described several times in ClinVar. Of note, the effect of this variant in splicing (causing skipping of exon 4) has been already characterized by using a splicing reporter minigene, as the authors mention in the discussion (Sanoguera-Miralles et al., Cancers (Basel) 2020 Dec 15;12(12):3771), diminishing the novelty of the finding. In my opinion, the only novelty of the paper is the functional assay performed for the p.Ser46Cys variant in RAD51D

Minor comments:

  • Figure S1 should be moved to the main manuscript, it would make it easier to follow the study design and different series analysed.
  • Some cases used for the discovery phase had been previously been tested using multi-gene panels. Were RAD51C&D not included in those panels?
  • LOH analysis results are all in supplementary information. They should be more clearly described and discussed in the main text.
  • As far as I understand, in the cDNA analysis performed on the c.705G>T; Lys235Asn variant, the authors have not ruled out whether there is full-length mRNA being transcribed from the variant allele. Confirming that only exon 4 skipped mRNA is being transcribed from the “mutant” allele is important for final classification of the variant.
  • I think it would be good to add a known pathogenic RAD51D variant as “positive” control in the functional assay performed on p.Ser46Cys in

Author Response

(The authors gave the same response as above.)
